# MODULARIZED MORPHING OF NEURAL NETWORKS

**Tao Wei**[†]
University at Buffalo
Buffalo, NY 14260
`taowei@buffalo.edu`

**Changhu Wang**
Microsoft Research
Beijing, China, 100080
`chw@microsoft.com`

**Chang Wen Chen**
University at Buffalo
Buffalo, NY 14260
`chencw@buffalo.edu`

## ABSTRACT

In this work we study the problem of network morphism, an effective learning scheme to morph a well-trained neural network to a new one with the network function completely preserved. Different from existing work where basic morphing types on the layer level were addressed, we target at the central problem of network morphism at a higher level, i.e., *how a convolutional layer can be morphed into an arbitrary module of a neural network*. To simplify the representation of a network, we abstract a module as a graph with blobs as vertices and convolutional layers as edges, based on which the morphing process is able to be formulated as a graph transformation problem. Two atomic morphing operations are introduced to compose the graphs, based on which modules are classified into two families, i.e., simple morphable modules and complex modules. We present practical morphing solutions for both of these two families, and prove that any reasonable module can be morphed from a single convolutional layer. Extensive experiments have been conducted based on the state-of-the-art ResNet on benchmark datasets, and the effectiveness of the proposed solution has been verified.

## 1 INTRODUCTION

Deep convolutional neural networks have continuously demonstrated their excellent performances on diverse computer vision problems. In image classification, the milestones of such networks can be roughly represented by LeNet (LeCun et al., 1989), AlexNet (Krizhevsky et al., 2012), VGG net (Simonyan & Zisserman, 2014), GoogLeNet (Szegedy et al., 2014), and ResNet (He et al., 2015), with networks becoming deeper and deeper. However, the architectures of these network are significantly altered and hence are not backward-compatible. Considering a life-long learning system, it is highly desired that the system is able to update itself from the original version established initially, and then evolve into a more powerful one, rather than re-learning a brand new one from scratch.

Network morphism (Wei et al., 2016) is an effective way towards such an ambitious goal. It can morph a well-trained network to a new one with the knowledge entirely inherited, and hence is able to update the original system to a compatible and more powerful one based on further training. Network morphism is also a performance booster and architecture explorer for convolutional neural networks, allowing us to quickly investigate new models with significantly less computational and human resources. However, the network morphism operations proposed in (Wei et al., 2016), including depth, width, and kernel size changes, are quite primitive and have been limited to the level of layer in a network. For practical applications where neural networks usually consist of dozens or even hundreds of layers, the morphing space would be too large for researchers to practically design the architectures of target morphed networks, when based on these primitive morphing operations only.

Different from previous work, we investigate in this research the network morphism from a higher level of viewpoint, and systematically study the central problem of network morphism on the module level, i.e., *whether and how a convolutional layer can be morphed into an arbitrary module*[1], where a module refers to a single-source, single-sink acyclic subnet of a neural network. With this

---

[†]Tao Wei performed this work while being an intern at Microsoft Research Asia.
[1]Although network morphism generally does not impose constraints on the architecture of the child network, in this work we limit the investigation to the expanding mode.

modularized network morphing, instead of morphing in the layer level where numerous variations exist in a deep neural network, we focus on the changes of basic modules of networks, and explore the morphing space in a more efficient way. The necessities for this study are two folds. First, we wish to explore the capability of the network morphism operations and obtain a theoretical upper bound for what we are able to do with this learning scheme. Second, modern state-of-the-art convolutional neural networks have been developed with modularized architectures (Szegedy et al., 2014; He et al., 2015), which stack the construction units following the same module design. It is highly desired that the morphing operations could be directly applied to these networks.

To study the morphing capability of network morphism and figure out the morphing process, we introduce a simplified graph-based representation for a module. Thus, the network morphing process can be formulated as a graph transformation process. In this representation, the module of a neural network is abstracted as a directed acyclic graph (DAG), with data blobs in the network represented as vertices and convolutional layers as edges. Furthermore, a vertex with more than one outdegree (or indegree) implicitly includes a split of multiple copies of blobs (or a joint of addition). Indeed, the proposed graph abstraction suffers from the problem of dimension compatibility of blobs, for different kernel filters may result in totally different blob dimensions. We solve this problem by extending the blob and filter dimensions from finite to infinite, and the convergence properties will also be carefully investigated.

Two atomic morphing operations are adopted as the basis for the proposed graph transformation, based on which a large family of modules can be transformed from a convolutional layer. This family of modules are called simple morphable modules in this work. A novel algorithm is proposed to identify the morphing steps by reducing the module into a single convolutional layer. For any module outside the simple morphable family, i.e., complex module, we first apply the same reduction process and reduce it to an irreducible module. A practical algorithm is then proposed to solve for the network morphism equation of the irreducible module. Therefore, we not only verify the morphability to an arbitrary module, but also provide a unified morphing solution. This demonstrates the generalization ability and thus practicality of this learning scheme.

Extensive experiments have been conducted based on ResNet (He et al., 2015) to show the effectiveness of the proposed morphing solution. With only 1.2x or less computation, the morphed network can achieve up to 25% relative performance improvement over the original ResNet. Such an improvement is significant in the sense that the morphed 20-layered network is able to achieve an error rate of 6.60% which is even better than a 110-layered ResNet (6.61%) on the CIFAR10 dataset, with only around $1/5$ of the computational cost. It is also exciting that the morphed 56-layered network is able to achieve 5.37% error rate, which is even lower than those of ResNet-110 (6.61%) and ResNet-164 (5.46%). The effectiveness of the proposed learning scheme has also been verified on the CIFAR100 and ImageNet datasets.

## 2 RELATED WORK

*Knowledge Transfer.* Network morphism originated from knowledge transferring for convolutional neural networks. Early attempts were only able to transfer partial knowledge of a well-trained network. For example, a series of model compression techniques (Bucilu et al., 2006; Ba & Caruana, 2014; Hinton et al., 2015; Romero et al., 2014) were proposed to fit a lighter network to predict the output of a heavier network. Pre-training (Simonyan & Zisserman, 2014) was adopted to pre-initialize certain layers of a deeper network with weights learned from a shallower network. However, network morphism requires the knowledge being fully transferred, and existing work includes Net2Net (Chen et al., 2015) and NetMorph (Wei et al., 2016). Net2Net achieved this goal by padding identity mapping layers into the neural network, while NetMorph decomposed a convolutional layer into two layers by deconvolution. Note that the network morphism operations in (Chen et al., 2015; Wei et al., 2016) are quite primitive and at a micro-scale layer level. In this research, we study the network morphism at a meso-scale module level, and in particular, we investigate its morphing capability.

*Modularized Network Architecture.* The evolution of convolutional neural networks has been from sequential to modularized. For example, LeNet (LeCun et al., 1998), AlexNet (Krizhevsky et al., 2012), and VGG net (Simonyan & Zisserman, 2014) are sequential networks, and their difference is primarily on the number of layers, which is 5, 8, and up to 19 respectively. However, recently

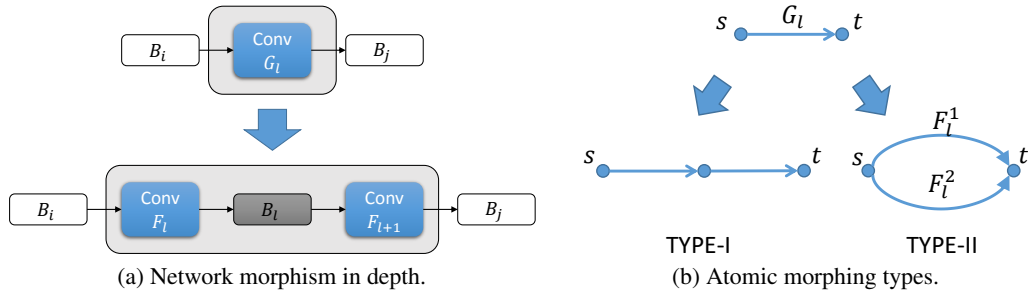

(a) Network morphism in depth.
(b) Atomic morphing types.

Figure 1: Illustration of atomic morphing types. (a) One convolutional layer is morphed into two convolutional layers; (b) `TYPE-I` and `TYPE-II` atomic morphing types.

proposed networks, such as GoogLeNet (Szegedy et al., 2014; 2015) and ResNet (He et al., 2015), follow a modularized architecture design, and have achieved the state-of-the-art performance. This is why we wish to study network morphism at the module level, so that its operations are able to directly apply to these modularized network architectures.

## 3 NETWORK MORPHISM VIA GRAPH ABSTRACTION

In this section, we present a systematic study on the capability of network morphism learning scheme. We shall verify that a convolutional layer is able to be morphed into any single-source, single-sink DAG subnet, named as a module here. We shall also present the corresponding morphing algorithms.

For simplicity, we first consider convolutional neural networks with only convolutional layers. All other layers, including the non-linearity and batch normalization layers, will be discussed later in this paper.

### 3.1 BACKGROUND AND BASIC NOTATIONS

For a 2D deep convolutional neural network (DCNN), as shown in Fig. 1a, the convolution is defined by:

$$B_j(c_j) = \sum_{c_i} B_i(c_i) * G_l(c_j, c_i), \tag{1}$$

where the blob $B_*$ is a 3D tensor of shape $(C_*, H_*, W_*)$ and the convolutional filter $G_l$ is a 4D tensor of shape $(C_j, C_i, K_l, K_l)$. In addition, $C_*$, $H_*$, and $W_*$ represent the number of channels, height and width of $B_*$. $K_l$ is the convolutional kernel size[2].

In a network morphism process, the convolutional layer $G_l$ in the parent network is morphed into two convolutional layers $F_l$ and $F_{l+1}$ (Fig. 1a), where the filters $F_l$ and $F_{l+1}$ are 4D tensors of shapes $(C_l, C_i, K_1, K_1)$ and $(C_j, C_l, K_2, K_2)$. This process should follow the morphism equation:

$$\tilde{G}_l(c_j, c_i) = \sum_{c_l} F_l(c_l, c_i) * F_{l+1}(c_j, c_l), \tag{2}$$

where $\tilde{G}_l$ is a zero-padded version of $G_l$ whose effective kernel size is $\tilde{K}_l = K_1 + K_2 - 1 \geq K_l$. (Wei et al., 2016) showed a sufficient condition for exact network morphism:

$$\max(C_l C_i K_1^2, C_j C_l K_2^2) \geq C_j C_i (K_1 + K_2 - 1)^2. \tag{3}$$

For simplicity, we shall denote equations (1) and (2) as $B_j = G_l \circledast B_i$ and $\tilde{G}_l = F_{l+1} \circledast F_l$, where $\circledast$ is a non-communicative multi-channel convolution operator. We can also rewrite equation (3) as $\max(|F_l|, |F_{l+1}|) \geq |\tilde{G}_l|$, where $|*|$ measures the size of the convolutional filter.

---

[2]Generally speaking, $G_l$ is a 4D tensor of shape $(C_j, C_i, K_l^H, K_l^W)$, where convolutional kernel sizes for blob height and width are not necessary to be the same. However, in order to simplify the notations, we assume that $K_l^H = K_l^W$, but the claims and theorems in this paper apply equally well when they are different.

## 3.2 ATOMIC NETWORK MORPHISM

We start with the simplest cases. Two *atomic morphing types* are considered, as shown in Fig. 1b: 1) a convolutional layer is morphed into two convolutional layers (TYPE-I); 2) a convolutional layer is morphed into two-way convolutional layers (TYPE-II). For the TYPE-I atomic morphing operation, equation (2) is satisfied, while For TYPE-II, the filter split is set to satisfy

$$G_l = F_l^1 + F_l^2. \tag{4}$$

In addition, for TYPE-II, at the source end, the blob is split with multiple copies; while at the sink end, the blobs are joined by addition.

## 3.3 GRAPH ABSTRACTION

To simplify the representation, we introduce the following graph abstraction for network morphism. For a convolutional neural network, we are able to abstract it as a graph, with the blobs represented by vertices, and convolutional layers by edges. Formally, a DCNN is represented as a DAG $M = (V, E)$, where $V = \{B_i\}_{i=1}^N$ are blobs and $E = \{e_l = (B_i, B_j)\}_{l=1}^L$ are convolutional layers. Each convolutional layer $e_l$ connects two blobs $B_i$ and $B_j$, and is associated with a convolutional filter $F_l$. Furthermore, in this graph, if $outdegree(B_i) > 1$, it implicitly means a split of multiple copies; and if $indegree(B_i) > 1$, it is a joint of addition.

Based on this abstraction, we formally introduce the following definition for *modular network morphism*:

**Definition 1.** Let $M_0 = (\{s, t\}, e_0)$ represent the graph with only a single edge $e_0$ that connects the source vertex $s$ and sink vertex $t$. $M = (V, E)$ represents any single-source, single-sink DAG with the same source vertex $s$ and the same sink vertex $t$. We call such an $M$ as a *module*. If there exists a process that we are able to morph $M_0$ to $M$, then we say that module $M$ is *morphable*, and the morphing process is called *modular network morphism*.

Hence, based on this abstraction, modular network morphism can be represented as a graph transformation problem. As shown in Fig. 2b, module (C) in Fig. 2a can be transformed from module $M_0$ by applying the illustrated network morphism operations.

For each modular network morphing, a modular network morphism equation is associated:

**Definition 2.** Each module essentially corresponds to a function from $s$ to $t$, which is called a *module function*. For a modular network morphism process from $M_0$ to $M$, the equation that guarantees the module function unchanged is called *modular network morphism equation*.

It is obvious that equations (2) and (4) are the modular network morphism equations for TYPE-I and TYPE-II atomic morphing types. In general, the modular network morphism equation for a module $M$ is able to be written as the sum of all convolutional filter compositions, in which each composition is actually a path from $s$ to $t$ in the module $M$. Let $\{(F_{p,1}, F_{p,2}, \cdots, F_{p,i_p}) : p = 1, \cdots, P$, and $i_p$ is the length of path $p\}$ be the set of all such paths represented by the convolutional filters. Then the modular network morphism equation for module $M$ can be written as

$$G_l = \sum_p F_{p,i_p} \circledast F_{p,i_p-1} \circledast \cdots \circledast F_{p,1}. \tag{5}$$

As an example illustrated in Fig. 2a, there are four paths in module (D), and its modular network morphism equation can be written as

$$G_l = F_5 \circledast F_1 + F_6 \circledast (F_3 \circledast F_1 + F_4 \circledast F_2) + F_7 \circledast F_2, \tag{6}$$

where $G_l$ is the convolutional filter associated with $e_0$ in module $M_0$.

## 3.4 THE COMPATIBILITY OF NETWORK MORPHISM EQUATION

One difficulty in this graph abstraction is in the dimensional compatibility of convolutional filters or blobs. For example, for the TYPE-II atomic morphing in Fig. 1b, we have to satisfy $G_l = F_l^1 + F_l^2$. Suppose that $G_l$ and $F_l^2$ are of shape $(64, 64, 3, 3)$, while $F_l^1$ is $(64, 64, 1, 1)$, they are actually not addable. Formally, we define the compatibility of modular network morphism equation as follows:

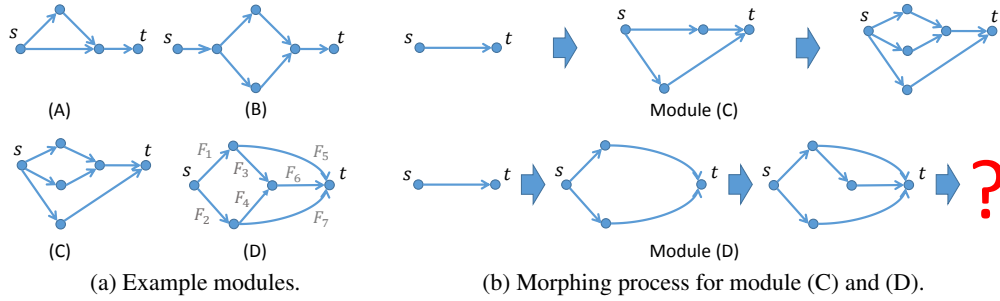

(a) Example modules.  (b) Morphing process for module (C) and (D).

Figure 2: Example modules and morphing processes. (a) Modules (A)-(C) are simple morphable, while (D) is not; (b) a morphing process for module (C), while for module (D), we are not able to find such a process.

**Definition 3.** The modular network morphism equation for a module $M$ is *compatible* if and only if the mathematical operators between the convolutional filters involved in this equation are well-defined.

In order to solve this compatibility problem, we need not to assume that blobs $\{B_i\}$ and filters $\{F_l\}$ are finite dimension tensors. Instead they are considered as infinite dimension tensors defined with a finite support[3], and we call this as an extended definition. An instant advantage when we adopt this extended definition is that we will no longer need to differentiate $G_l$ and $\tilde{G}_l$ in equation (2), since $\tilde{G}_l$ is simply a zero-padded version of $G_l$.

**Lemma 4.** *The operations $+$ and $\circledast$ are well-defined for the modular network morphism equation. Namely, if $F^1$ and $F^2$ are infinite dimension 4D tensors with finite support, let $G = F^1 + F^2$ and $H = F^2 \circledast F^1$, then both $G$ and $H$ are uniquely defined and also have finite support.*

*Sketch of Proof.* It is quite obvious that this lemma holds for the operator $+$. For the operator $\circledast$, if we have this extended definition, the sum in equation (2) will become infinite over the index $c_l$. It is straightforward to show that this infinite sum converges, and also that $H$ is finitely supported with respect to the indices $c_j$ and $c_i$. Hence $H$ has finite support. $\square$

As a corollary, we have:

**Corollary 5.** *The modular network morphism equation for any module $M$ is always compatible if the filters involved in $M$ are considered as infinite dimension tensors with finite support.*

### 3.5 SIMPLE MORPHABLE MODULES

In this section, we introduce a large family of modules, i.e, simple morphable modules, and then provide their morphing solutions. We first introduce the following definition:

**Definition 6.** A module $M$ is *simple morphable* if and only if it is able to be morphed with only combinations of atomic morphing operations.

Several example modules are shown in Fig. 2a. It is obvious that modules (A)-(C) are simple morphable, and the morphing process for module (C) is also illustrated in Fig. 2b.

For a simple morphable module $M$, we are able to identity a morphing sequence from $M_0$ to $M$. The algorithm is illustrated in Algorithm 1. The core idea is to use the reverse operations of atomic morphing types to reduce $M$ to $M_0$. Hence, the morphing process is just the reverse of the reduction process. In Algorithm 1, we use a four-element tuple $(M, e_1, \{e_2, e_3\}, type)$ to represent the process of morphing edge $e_1$ in module $M$ to $\{e_2, e_3\}$ using TYPE-<TYPE> atomic operation. Two auxiliary functions CHECKTYPEI and CHECKTYPEII are further introduced. Both

---

[3]A support of a function is defined as the set of points where the function value is non-zero, i.e., $support(f) = \{x | f(x) \neq 0\}$.

---

**Algorithm 1** Algorithm for Simple Morphable Modules

---
**Input:** $M_0$; a simple morphable module $M$
**Output:** The morphing sequence $Q$ that morphs $M_0$ to $M$ using atomic morphing operations
$Q = \emptyset$
**while** $M \neq M_0$ **do**
 **while** CHECKTYPEI($M$) is not FALSE **do**
 // Let $(M_{temp}, e_1, \{e_2, e_3\}, type)$ be the return value of CHECKTYPEI($M$)
 $Q.prepend((M_{temp}, e_1, \{e_2, e_3\}, type))$ and $M \leftarrow M_{temp}$
 **end while**
 **while** CHECKTYPEII($M$) is not FALSE **do**
 // Let $(M_{temp}, e_1, \{e_2, e_3\}, type)$ be the return value of CHECKTYPEII($M$)
 $Q.prepend((M_{temp}, e_1, \{e_2, e_3\}, type))$ and $M \leftarrow M_{temp}$
 **end while**
**end while**

---

**Algorithm 2** Algorithm for Irreducible Modules

---
**Input:** $G_l$; an irreducible module $M$
**Output:** Convolutional filters $\{F_i\}_{i=1}^n$ of $M$
Initialize $\{F_i\}_{i=1}^n$ with random noise.
Calculate the effective kernel size of $M$, expand $G_l$ to $\tilde{G}_l$ by padding zeros.
**repeat**
 **for** $j = 1$ to $n$ **do**
 Fix $\{F_i : i \neq j\}$, and calculate $F_j = deconv(\tilde{G}_l, \{F_i : i \neq j\})$
 Calculate loss $l = \|\tilde{G}_l - conv(\{F_i\}_{i=1}^n)\|^2$
 **end for**
**until** $l = 0$ or $maxIter$ is reached

---

of them return either FALSE if there is no such atomic sub-module in $M$, or a morphing tuple $(M, e_1, \{e_2, e_3\}, type)$ if there is. The algorithm of CHECKTYPEI only needs to find a vertex satisfying $indegree(B_i) = outdegree(B_i) = 1$, while CHECKTYPEII looks for the matrix elements $> 1$ in the adjacent matrix representation of module $M$.

Is there a module not simple morphable? The answer is yes, and an example is the module (D) in Fig. 2a. A simple try does not work as shown in Fig. 2b. In fact, we have the following proposition:

**Proposition 7.** *Module (D) in Fig. 2a is not simple morphable.*

*Sketch of Proof.* A simple morphable module $M$ is always able to be reverted back to $M_0$. However, for module (D) in Fig. 2a, both CHECKTYPEI and CHECKTYPEII return FALSE. □

## 3.6 MODULAR NETWORK MORPHISM THEOREM

For a module that is not simple morphable, which is called a *complex module*, we are able to apply Algorithm 1 to reduce it to an irreducible module $M$ first. For $M$, we propose Algorithm 2 to solve the modular network morphism equation. The core idea of this algorithm is that, if only one convolutional filter is allowed to change with all others fixed, the modular network morphism equation will reduce to a linear system. The following argument guarantees the correctness of Algorithm 2.

*Correctness of Algorithm 2.* Let $G_l$ and $\{F_i\}_{i=1}^n$ be the convolutional filter(s) associated with $M_0$ and $M$. We further assume that one of $\{F_i\}$, e.g., $F_j$, is larger or equal to $\tilde{G}_l$, where $\tilde{G}_l$ is the zero-padded version of $G_l$ (this assumption is a strong condition in the expanding mode). The module network morphism equation for $M$ can be written as

$$\tilde{G}_l = C_1 \circledast F_j \circledast C_2 + C_3, \tag{7}$$

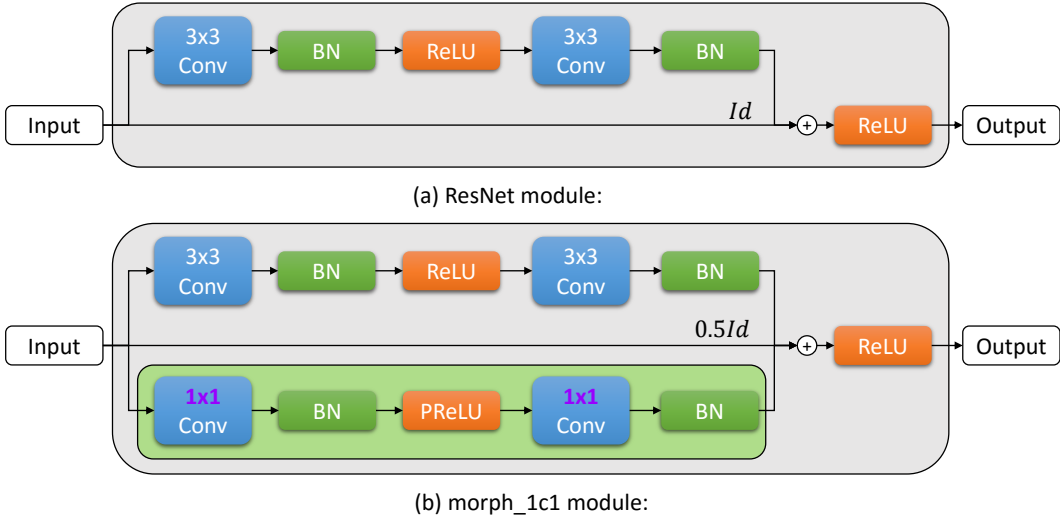

(a) ResNet module:

(b) morph_1c1 module:

Figure 3: Detailed architectures of the ResNet module and the `morph_1c1` module.

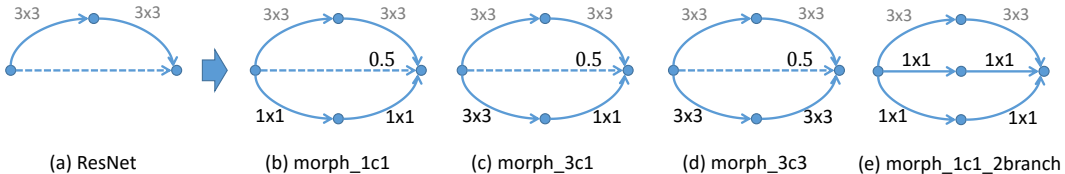

(a) ResNet    (b) morph_1c1    (c) morph_3c1    (d) morph_3c3    (e) morph_1c1_2branch

Figure 4: Sample modules adopted in the proposed experiments. (a) and (b) are the graph abstractions of modules illustrated in Fig. 3(a) and (b).

where $C_1$, $C_2$, and $C_3$ are composed of other filters $\{F_i : i \neq j\}$. It can be checked that equation (7) is a linear system with $|\tilde{G}_l|$ constraints and $|F_j|$ free variables. Since we have $|F_j| \geq |\tilde{G}_l|$, the system is non-deterministic and hence solvable as random matrices are rarely inconsistent. □

For a general module $M$, whether simple morphable or not, we apply Algorithm 1 to reduce $M$ to an irreducible module $M'$, and then apply Algorithm 2 to $M'$. Hence we have the following theorem:

**Theorem 8.** *A convolutional layer can be morphed to any module (any single-source, single-sink DAG subnet).*

This theorem answers the core question of network morphism, and provides a theoretical upper bound for the capability of this learning scheme.

### 3.7    NON-LINEARITY AND BATCH NORMALIZATION IN MODULAR NETWORK MORPHISM

Besides the convolutional layers, a neural network module typically also involves non-linearity layers and batch normalization layers, as illustrated in Fig. 3. In this section, we shall describe how do we handle these layers for modular network morphism.

For the non-linear activation layers, we adopt the solution proposed in (Wei et al., 2016). Instead of directly applying the non-linear activations, we are using their parametric forms. Let $\varphi$ be any non-linear activation function, and its parametric form is defined to be

$$P\text{-}\varphi = \{\varphi^a\}|_{a \in [0,1]} = \{(1-a) \cdot \varphi + a\varphi_{id}\}|_{a \in [0,1]}. \tag{8}$$

The shapes of the parametric form of the non-linear activation $\varphi$ is controlled by the parameter $a$. When $a$ is initialized ($a = 1$), the parametric form is equivalent to an identity function, and when the value of $a$ has been learned, the parametric form will become a non-linear activation. In Fig. 3b, the non-linear activation for the morphing process is annotated as PReLU to differentiate itself with the

Table 1: Experimental results of networks morphed from ResNet-20, ResNet-56, and ResNet-110 on the CIFAR10 dataset. Results annotated with [†] are from (He et al., 2015).

| Net Arch. | Intermediate Phases | Error | Abs. Perf. Improv. | Rel. Perf. Improv. | #Params. (MB) | #Params. Rel. | FLOP (million) | Rel. FLOP |
|---|---|---|---|---|---|---|---|---|
| resnet20[†] | - | 8.75% | - | - | 1.048 | 1x | 40.8 | 1x |
| morph20_1c1 | - | 7.35% | 1.40% | 16.0% | 1.138 | 1.09x | 44.0 | 1.08x |
| morph20_3c1 | - | 7.10% | 1.65% | 18.9% | 1.466 | 1.40x | 56.5 | 1.38x |
| | 1c1 | 6.83% | 1.92% | 21.9% | | | | |
| morph20_3c3 | - | 6.97% | 1.78% | 20.3% | 1.794 | 1.71x | 69.1 | 1.69x |
| | 1c1,3c1 | 6.66% | 2.09% | 23.9% | | | | |
| morph20_1c1_2branch | - | 7.26% | 1.49% | 17.0% | 1.227 | 1.17x | 47.1 | 1.15x |
| | 1c1,half | **6.60%** | **2.15%** | **24.6%** | | | | |
| resnet56[†] | - | 6.97% | - | - | 3.289 | 1x | 125.7 | 1x |
| morph56_1c1_half | - | 5.68% | 1.29% | 18.5% | 3.468 | 1.05x | 132.0 | 1.05x |
| morph56_1c1 | - | 5.94% | 1.03% | 14.8% | 3.647 | 1.11x | 138.3 | 1.10x |
| | 1c1_half | **5.37%** | **1.60%** | **23.0%** | | | | |
| resnet110[†] | - | 6.61%±0.16 | - | - | 6.649 | 1x | 253.1 | 1x |
| morph110_1c1_half | - | 5.74% | 0.87% | 13.2% | 7.053 | 1.06x | 267.3 | 1.06x |
| morph110_1c1 | - | 5.93% | 0.68% | 10.3% | 7.412 | 1.11x | 279.9 | 1.11x |
| | 1c1_half | **5.50%** | **1.11%** | **16.8%** | | | | |

Table 2: Comparison results between learning from morphing and learning from scratch for the same network architectures on the CIFAR10 dataset.

| Net Arch. | Error (scratch) | Error (morph) | Abs. Perf. Improv. | Rel. Perf. Improv. |
|---|---|---|---|---|
| morph20_1c1 | 8.01% | 7.35% | 0.66% | 8.2% |
| morph20_1c1_2branch | 7.90% | 6.60% | 1.30% | 16.5% |
| morph56_1c1 | 7.37% | 5.37% | 2.00% | 27.1% |
| morph110_1c1 | 8.16% | 5.50% | 2.66% | 32.6% |

other ReLU activations. In the proposed experiments, for simplicity, all ReLUs are replaced with PReLUs.

The batch normalization layers (Ioffe & Szegedy, 2015) can be represented as

$$newdata = \frac{data - mean}{\sqrt{var + eps}} \cdot gamma + beta. \tag{9}$$

It is obvious that if we set $gamma = \sqrt{var + eps}$ and $beta = mean$, then a batch normalization layer is reduced to an identity mapping layer, and hence it can be inserted anywhere in the network. Although it is possible to calculate the values of $gamma$ and $beta$ from the training data, in this research, we adopt another simpler approach by setting $gamma = 1$ and $beta = 0$. In fact, the value of $gamma$ can be set to any nonzero number, since the scale is then normalized by the latter batch normalization layer (lower right one in Fig. 3b). Mathematically and strictly speaking, when we set $gamma = 0$, the network function is actually changed. However, since the morphed filters for the convolutional layers are roughly randomized, even though the $mean$ of $data$ is not strictly zero, it is still approximately zero. Plus with the fact that the data is then normalized by the latter batch normalization layer, such small perturbation for the network function change can be neglected. In the proposed experiments, only statistical variances in performance are observed for the morphed network when we adopt setting $gamma$ to zero. The reason we prefer such an approach to using the training data is that it is easier to implement and also yields slightly better results when we continue to train the morphed network.

## 4 EXPERIMENTAL RESULTS

In this section, we report the results of the proposed morphing algorithms based on current state-of-the-art ResNet (He et al., 2015), which is the winner of 2015 ImageNet classification task.

Table 3: Experimental results of networks morphed from ResNet-20, ResNet-56, and ResNet-110 on the CIFAR100 dataset.

| Net Arch. | Intermediate Phases | Error | Abs. Perf. Improv. | Rel. Perf. Improv. | #Params. (MB) | #Params. Rel. | FLOP (million) | Rel. FLOP |
|---|---|---|---|---|---|---|---|---|
| resnet20 | - | 32.82% | - | - | 1.070 | 1x | 40.8 | 1x |
| morph20_1c1 | - | 31.70% | 1.12% | 3.4% | 1.160 | 1.08x | 44.0 | 1.08x |
| resnet56 | - | 29.83% | - | - | 3.311 | 1x | 125.8 | 1x |
| morph56_1c1 | 1c1_half | 27.52% | 2.31% | 7.7% | 3.670 | 1.11x | 138.3 | 1.10x |
| resnet110 | - | 28.46% | - | - | 6.672 | 1x | 253.2 | 1x |
| morph110_1c1 | 1c1_half | 26.81% | 1.65% | 5.8% | 7.434 | 1.11x | 279.9 | 1.11x |

Table 4: Comparison results between learning from morphing and learning from scratch for the same network architectures on the CIFAR100 dataset.

| Net Arch. | Error (scratch) | Error (morph) | Abs. Perf. Improv. | Rel. Perf. Improv. |
|---|---|---|---|---|
| morph20_1c1 | 33.63% | 31.70% | 1.93% | 5.7% |
| morph56_1c1 | 32.58% | 27.52% | 5.06% | 15.5% |
| morph110_1c1 | 31.94% | 26.81% | 5.13% | 16.1% |

## 4.1 NETWORK ARCHITECTURES OF MODULAR NETWORK MORPHISM

We first introduce the network architectures used in the proposed experiments. Fig. 3a shows the module template in the design of ResNet (He et al., 2015), which is actually a simple morphable two-way module. The first path consists of two convolutional layers, and the second path is a shortcut connection of identity mapping. The architecture of the ResNet module can be abstracted as the graph in Fig. 4a. For the morphed networks, we first split the identity mapping layer in the ResNet module into two layers with a scaling factor of 0.5. Then each of the scaled identity mapping layers is able to be further morphed into two convolutional layers. Fig. 3b illustrates the case with only one scaled identity mapping layer morphed into two convolutional layers, and its equivalent graph abstraction is shown in Fig. 4b. To differentiate network architectures adopted in this research, the notation morph_<k1>c<k2> is introduced, where k1 and k2 are kernel sizes in the morphed network. If both of scaled identity mapping branches are morphed, we append a suffix of '_2branch'. Some examples of morphed modules are illustrated in Fig. 4. We also use the suffix '_half' to indicate that only one half (odd-indexed) of the modules are morphed, and the other half are left as original ResNet modules.

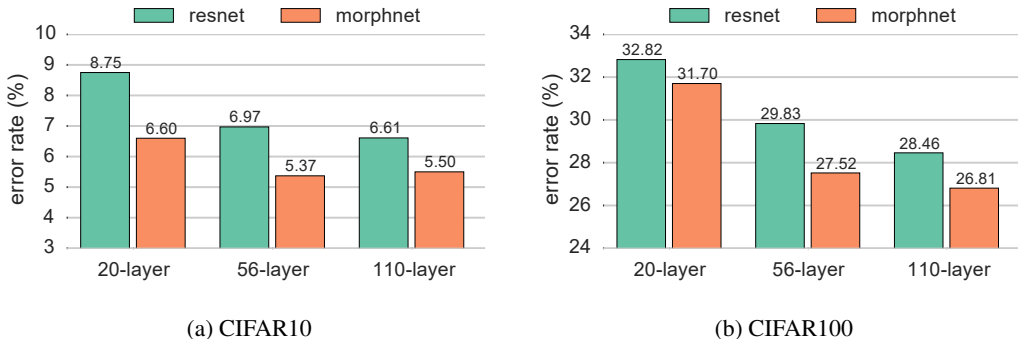

(a) CIFAR10

(b) CIFAR100

Figure 5: Comparison results of ResNet and morphed networks on the CIFAR10 and CIFAR100 datasets.

## 4.2 EXPERIMENTAL RESULTS ON THE CIFAR10 DATASET

CIFAR10 (Krizhevsky & Hinton, 2009) is a benchmark dataset on image classification and neural network investigation. It consists of $32 \times 32$ color images in 10 categories, with 50,000 training images and 10,000 testing images. In the training process, we follow the same setup as in (He et al., 2015). We use a decay of 0.0001 and a momentum of 0.9. We adopt the simple data augmentation with a pad of 4 pixels on each side of the original image. A $32 \times 32$ view is randomly cropped from the padded image and a random horizontal flip is optionally applied.

Table 1 shows the results of different networks morphed from ResNet (He et al., 2015). Notice that it is very challenging to further improve the performance, for ResNet has already boosted the number to a very high level. E.g., ResNet (He et al., 2015) made only 0.36% performance improvement by extending the model from 56 to 110 layers (Table 1). From Table 1 we can see that, with only 1.2x or less computational cost, the morphed networks achieved 2.15%, 1.60%, 1.11% performance improvements over the original ResNet-20, ResNet-56, and ResNet-110 respectively. Notice that the relative performance improvement can be up to 25%. Table 1 also compares the number of parameters of the original network architectures and the ones after morphing. As can be seen, the morphed ones only have a little more parameters than the original ones, typically less than 1.2x.

Except for large error rate reduction achieved by the morphed network, one exciting indication from Table 1 is that the morphed 20-layered network `morph20_3c3` is able to achieve slightly lower error rate than the 110-layered ResNet (6.60% vs 6.61%), and its computational cost is actually less than $1/5$ of the latter one. Similar results have also been observed from the morphed 56-layered network. It is able to achieve a 5.37% error rate, which is even lower than those of ResNet-110 (6.61%) and ResNet-164 (5.46%) (He et al., 2016). These results are also illustrated in Fig. 5(a).

Several different architectures of the morphed networks were also explored, as illustrated in Fig. 4 and Table 1. First, when the kernel sizes were expanded from $1 \times 1$ to $3 \times 3$, the morphed networks (`morph20_3c1` and `morph20_3c3`) achieved better performances. Similar results were reported in (Simonyan & Zisserman, 2014) (Table 1 for models C and D). However, because the morphed networks almost double the computational cost, we did not adopt this approach. Second, we also tried to morph the other scaled identity mapping layer into two convolutional layers (`morph20_1c1_2branch`), the error rate was further lowered for the 20-layered network. However, for the 56-layered and 110-layered networks, this strategy did not yield better results.

We also found that the morphed network learned with multiple phases could achieve a lower error rate than that learned with single phase. For example, the networks `morph20_3c1` and `morph20_3c3` learned with intermediate phases achieved better results in Table 1. This is quite reasonable as it divides the optimization problem into sequential phases, and thus is possible to avoid being trapped into a local minimum to some extent. Inspired by this observation, we then used a `1c1_half` network as an intermediate phase for the `morph56_1c1` and `morph110_1c1` networks, and better results have been achieved.

We compared the proposed learning scheme against learning from scratch for the networks with the same architectures. These results are illustrated in Table 2. As can be seen, networks learned by morphing is able to achieve up to 2.66% absolute performance improvement and 32.6% relative performance improvement comparing against learning from scratch for the `morph110_1c1` network architecture. These results are quite reasonable as when networks are learned by the proposed morphing scheme, they have already been regularized and shall have lower probability to be trapped into a bad-performing local minimum in the continual training process than the learning from scratch scheme. One may also notice that, `morph110_1c1` actually performed worse than `resnet110` when learned from scratch. This is because the network architecture `morph_1c1` is proposed for morphing, and the identity shortcut connection is scaled with a factor of 0.5. It was also reported that residual networks with a constant scaling factor of 0.5 actually led to a worse performance in (He et al., 2016) (Table 1), while this performance degradation problem could be avoided by the proposed morphing scheme.

Finally, it is worth noting that another advantage of the proposed learning scheme against the learning from scratch scheme is on model exploration. One can quickly check whether a morphed architecture deserves further exploration by continuing to train the morphed network in a finer learning rate (e.g. 1e-5), to see if the performance is improved. Hence, one does not have to wait for days or even months of training time to tell whether the new network architecture is able to achieve a

Table 5: Experimental results of networks morphed from ResNet-18 on the ImageNet dataset.

| Net Arch. | Eval. Mode | Top-1 Error | Abs. Perf. Improv. | Rel. Perf. Improv. | FLOP (billion) | Rel. FLOP |
|---|---|---|---|---|---|---|
| resnet18 | 1-view | 32.56% | - | - | 1.814 | 1x |
| | 10-view | 30.86% | - | - | | |
| morph18_1c1 | 1-view | 31.69% | 0.87% | 2.7% | 1.917 | 1.06x |
| | 10-view | 29.90% | 0.96% | 3.1% | | |
| resnet34 | 1-view | 29.08% | - | - | 3.664 | 1x |
| | 10-view | 27.32% | - | - | | |
| morph34_1c1 | 1-view | 27.90% | 1.18% | 4.1% | 3.972 | 1.08x |
| | 10-view | 26.20% | 1.12% | 4.1% | | |

better performance. This could save human time for deciding which network architecture is worth for exploring.

## 4.3 EXPERIMENTAL RESULTS ON THE CIFAR100 DATASET

CIFAR100 (Krizhevsky & Hinton, 2009) is another benchmark dataset for tiny images that consists of 100 categories. There are 500 training images and 100 testing images per category. The proposed experiments on CIFAR100 follows the same setup as in the experiments on CIFAR10. The experimental results are illustrated in Table 3 and Fig. 5(b). As shown, the performance improvement is also significant: with only around 1.1x computational cost, the absolute performance improvement can be up to 2% and the relative performance improvement can be up to 8%. For the morphed 56-layered network, it also achieves better performance than the 110-layered ResNet (27.52% vs 28.46%), and with only around one half of the computation. Table 4 also compares the proposed learning scheme against learning from scratch. More than 5% absolute performance improvement and around 16% relative performance improvement were achieved.

## 4.4 EXPERIMENTAL RESULTS ON THE IMAGENET DATASET

We also evaluate the proposed scheme on the ImageNet dataset (Russakovsky et al., 2014). This dataset consists of 1,000 object categories, with 1.28 million training images and 50K validation images. For the training process, we use a decay of 0.0001 and a momentum of 0.9. The image is resized to guarantee its shorter edge is randomly sampled from [256,480] for scale augmentation. A 224× 224 patch or its horizontal flip is randomly cropped from the re-

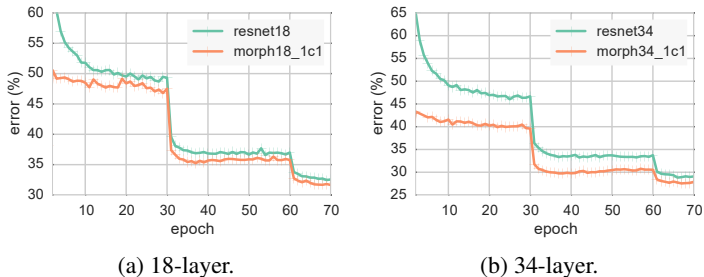

(a) 18-layer.    (b) 34-layer.

Figure 6: Evaluation errors on the ImageNet dataset.

sized image, with the image data per-channel normalized. We train the networks using SGD with a batch size of 256. The learning rate starts from 0.1 and is decreased with a factor of 0.1 for every 30 epochs. The networks are trained for a total of 70 epochs.

The comparison results of the morphed and original ResNets for both 18-layer and 34-layer networks are illustrated in Table 5 and Fig. 6. As shown in Table 5, morph18_1c1 and morph34_1c1 are able to achieve lower error rates than ResNet-18 and ResNet-34 respectively, and the absolute performance improvements can be up to 1.2%. We also draw the evaluation error curves in Fig. 6, which shows that the morphed networks morph18_1c1 and morph34_1c1 are much more effective than the original ResNet-18 and ResNet-34 respectively.

## 5    CONCLUSIONS

This paper presented a systematic study on the problem of network morphism at a higher level, and tried to answer the central question of such learning scheme, i.e., whether and how a convolutional layer can be morphed into an arbitrary module. To facilitate the study, we abstracted a modular network as a graph, and formulated the process of network morphism as a graph transformation process. Based on this formulation, both simple morphable modules and complex modules have been defined and corresponding morphing algorithms have been proposed. We have shown that a convolutional layer can be morphed into any module of a network. We have also carried out experiments to illustrate how to achieve a better performing model based on the state-of-the-art ResNet with minimal extra computational cost on benchmark datasets. The experimental results have demonstrated the effectiveness of the proposed morphing approach.

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
