# Peer review of "Modularized Morphing of Neural Networks"

_ICLR 2017 — rejected_

[Official Review · AnonReviewer3 · rating 5 · confidence 4 · 15 Dec 2016]
**Interesting incremental approach for exploring network hierarchies, missing crucial experimental evidence.**

The paper presents an interesting incremental approach for exploring new convolutional network hierarchies in an incremental manner after a baseline network has reached a good recognition performance.

The experiments are presented for the CIFAR-100 and ImageNet benchmarks by morphing various ResNet models into better performing models with somewhat more computation.

Although the baselines are less strong than those presented in the literature, the paper claims significant error reduction for both ImageNet and CIFAR-100.

The main idea of the paper is to rewrite convolutions into multiple convolutions while expanding the number of filters. It is quite unexpected that this approach yields any improvements over the baseline model at all.

However, for some of the basic tenets of network morphing, experimental evidence is not given in the paper. Here are some fundamental questions raised by the paper:
- How does the quality of morphed networks compares to those with the same topology trained from scratch?
- How does the incremental training time after morphing relate to that of the network trained from scratch?
- Where is the extra computational cost of the morphed networks come from?
- Why is the quality of the baseline ResNet models lag behind those that are reported in the literature and github? (E.g. the github ResNet-101 model is supposed to have 6.1% top-5 recall vs 6.6 reported in the paper)
More evidence for the first three points would be necessary to evaluate the validity of the claims of the paper.

The paper is written reasonably well and can be understood quite well, but the missing evidence and weaker baselines make it looks somewhat less convincing. 
I would be inclined to revise up the score if a more experimental evidence were given for the main message of the paper (see the points above).

[Official Review · AnonReviewer2 · rating 7 · confidence 4 · 21 Dec 2016 (modified: 15 Jan 2017)]

The paper proposes a methodology for morphing a trained network to different architecture without having to retrain from scratch.
The manuscript reads well and the description is easy to follow.
However, the results are not very convincing as the selected baselines are considerably far from the state of the art.
The paper should include comparisons with state of the art, for example wide residual networks.
Tables should also report number of parameters for each architecture, this would help fair comparison.

[Official Review · AnonReviewer1 · rating 7 · confidence 5 · 28 Dec 2016]
**Modularized Morphing of Neural Networks**

This paper presents works on neural network / CNN architecture morphing.
Results are not reported on ImageNet larger ResNet and new network architecture such as Xception and DenseNet - which are maybe too new! Also most results are reported in small datasets and network, which do not offer confidence in the usability in production systems.
My biggest issue is that computational time and effort for these techniques is not mentioned in detail. We always want to be able to quantify the extra effort of understanding and using a new technique, especially if the results are minor.

[Official Review · AnonReviewer4 · rating 6 · confidence 5 · 02 Jan 2017]
**review for Modularized Morphing of Neural Networks**

This paper studies knowledge transfer problem from small capacity network to bigger one. This is a follow-up work of Net2Net (ICLR 2015) and NetMorph(ICML 2016).  
Comments
- 1) This paper studies macroscopic problem, with the morphing process composed by multiple atomic operations. While the atomic operations are proposed in Net2Net and NetMorph, there has not been study of the general modularized process principally. Thus this paper asks a novel question.
- 2) The solution by composing multiple atomic transformations seems to be quite reasonable.
- 3) In the “related work” section, it is better to change “network morphism” to “knowledge transfer” or in the subsection title, most of these works are known as knowledge transfer and it helps to connect to the existing works.
- 4) The author shows experiments on variants of ResNet. While the experiment shows that initializing from ResNet gives better error rate than the ones trained from scratch, it is unclear what the source This paper studies knowledge transfer problem from small capacity network to bigger one. This is a follow-up work of Net2Net (ICLR 2015) and NetMorph(ICML 2016).  
Comments
- 1) This paper studies macroscopic problem, with the morphing process composed by multiple atomic operations. While the atomic operations are proposed in Net2Net and NetMorph, there has not been study of the general modularized process principally. Thus this paper asks a novel question.
- 2) The solution by composing multiple atomic transformations seems to be quite reasonable.
- 3) In the “related work” section, it is better to change “network morphism” to “knowledge transfer” or in the subsection title, most of these works are known as knowledge transfer and it helps to connect to the existing works.
- 4) The author shows experiments on variants of ResNet. While the experiment shows that initializing from ResNet gives better error rate than the ones trained from scratch, it is unclear what the source is.
- 5) One major advantage of this type of knowledge transfer (Net2Net, NetMorph) is to speedup training and model exploration. There seems to be no experiments demonstrate such advantage (possibly due to the lose initialization of BatchNorm). This is the major drawback of this paper.
-6)  The method proposed by the author can in principle do quite complicated transformation, e.g. transform  an entire resnet from a single conv layer, the experiment only consists of simple module transformations, which in some way can be covered by atomic operations. It would be more interesting to see what the results of more complicated transformations are (even if they are not as effective). 

In summary, this paper studies a novel problem of knowledge transfer in a macroscopic level. The method could be of interest to the ICLR community. The experiments should be improved (comment 5) to make the results more convincing and practically useful and I strongly encourage the authors to do so.

[Final Decision · Program Chairs · 06 Feb 2017]
**ICLR committee final decision**

The paper investigates the problem of morphing one convolutional network into another with application to exploring the model space (starting from a pre-trained baseline model). The resulting morphed models perform better than the baseline, albeit at the cost of more parameters and training. Importantly, it has not been demonstrated that morphing leads to training time speed-up, which is an important factor to consider when exploring new architectures starting from pre-trained models. Still, the presented approach and the experiments would be of interest to the community, so I recommend the paper for workshop presentation.